# Evaluating Explainable Artificial Intelligence (XAI) techniques in chest radiology imaging through a human-centered Lens

Izegbua E. Ihongbe[1], Shereen Fouad[1]*, Taha F. Mahmoud[2], Arvind Rajasekaran[3], Bahadar Bhatia[3,4]

1 School of Computer Science and Digital Technologies, Aston University, Birmingham, United Kingdom, 2 Medical Imaging Department, University Hospital of Sharjah, Sharjah, United Arab Emirates, 3 Sandwell And West Birmingham Hospitals NHS Trust, Birmingham, United Kingdom, 4 University of Leicester, Leicester, United Kingdom

☯ These authors contributed equally to this work.
* S.fouad@aston.ac.uk

**Data Availability Statement:** The data underlying the results presented in the study are available from: (1) Kermany, D., Zhang, K. and Goldbaum,

## Abstract

The field of radiology imaging has experienced a remarkable increase in using of deep learning (DL) algorithms to support diagnostic and treatment decisions. This rise has led to the development of Explainable AI (XAI) system to improve the transparency and trust of complex DL methods. However, XAI systems face challenges in gaining acceptance within the healthcare sector, mainly due to technical hurdles in utilizing these systems in practice and the lack of human-centered evaluation/validation. In this study, we focus on visual XAI systems applied to DL-enabled diagnostic system in chest radiography. In particular, we conduct a user study to evaluate two prominent visual XAI techniques from the human perspective. To this end, we created two clinical scenarios for diagnosing pneumonia and COVID-19 using DL techniques applied to chest X-ray and CT scans. The achieved accuracy rates were 90% for pneumonia and 98% for COVID-19. Subsequently, we employed two well-known XAI methods, Grad-CAM (Gradient-weighted Class Activation Mapping) and LIME (Local Interpretable Model-agnostic Explanations), to generate visual explanations elucidating the AI decision-making process. The visual explainability results were shared through a user study, undergoing evaluation by medical professionals in terms of clinical relevance, coherency, and user trust. In general, participants expressed a positive perception of the use of XAI systems in chest radiography. However, there was a noticeable lack of awareness regarding their value and practical aspects. Regarding preferences, Grad-CAM showed superior performance over LIME in terms of coherency and trust, although concerns were raised about its clinical usability. Our findings highlight key user-driven explainability requirements, emphasizing the importance of multi-modal explainability and the necessity to increase awareness of XAI systems among medical practitioners. Inclusive design was also identified as a crucial need to ensure better alignment of these systems with user needs.

M., 2018. Labeled optical coherence tomography (oct) and chest x-ray images for classification. Mendeley data, 2(2), p.651.). https://www.kaggle.com/datasets/paultimothymooney/chest-xray-pneumonia (2) Soares Eduardo, Angelov P. CT scans collected from real patients in hospitals from Sao Paulo, Brazil, A large dataset of CT scans for SARS-CoV-2 (COVID-19) identification. 2020. https://www.kaggle.com/datasets/plameneduardo/sarscov2-ctscan-dataset (3) The author-generated an open access code on which the manuscript is based, has been provided as a supporting information - S1 Supporting Information. Colaboratory Python code for clinical case study 1 - using Chest X-ray Images - (https://colab.research.google.com/drive/1v7RSS-_Prgujr-BrAGeDR_vygX_Tf-7r?usp=sharing) - S2 Supporting Information, Colaboratory Python code for clinical case study 2 - using Chest CT Images (https://colab.research.google.com/drive/1Y1wjd9-sKLD6MaZDw4QVleSfAV22Ldb4?usp=sharing) The code is shred in a way that follows best practice and facilitates reproducibility and reuse.

**Funding:** The author(s) received no specific funding for this work.

**Competing interests:** The authors have declared that no competing interests exist.

## 1 Introduction

Respiratory diseases, such as pneumonia and COVID-19, have had a profound impact on global health. According to the World Health Organization, they rank as one of the leading global causes of mortality, resulting in the deaths of 3.2 million individuals annually and accounting for 81.7% of fatalities linked to chronic respiratory diseases [1]. These diseases significantly burden healthcare systems due to the costs associated with hospitalisation and treatment. Effective prevention, early diagnosis, and management are critical not only for individual well-being but also for reducing the strain on healthcare systems. In recent years, deep learning (DL) methodologies, particularly Convolutional Neural Networks (CNNs), have emerged as powerful tools for automatically detecting chest diseases using medical images like CT scans and X-rays [2–4]. DL, inspired by the human brain, enables computers to learn from extensive data using artificial neural networks. These methods have become the foundation of various Computer Assisted Diagnostic systems due to their remarkable ability to detect medical image abnormalities with high accuracy, including COVID-19 and pneumonia diseases (e.g. [5, 6]). However, the "black-box" nature of DL methods hinders their social acceptance and use in real-life clinical settings. This limitation arises from their inability to explain the rationale behind AI-based decisions, particularly in medical diagnosis, where model transparency and interpretability are pivotal for clinical trust and acceptance. There is a growing demand, especially among the medical community, to enhance the interpretability of DL techniques applied to medical images, increasingly fueled by emerging regulations concerning AI privacy, fairness, and awareness (e.g. European general data protection regulation(GDPR) [7]).

The concept of Explainable AI (XAI) focuses on implementing transparency and interpretability in black-box machine learning and deep learning methods [8, 9]. Recent XAI methods, particularly post-modeling approaches, have attracted significant attention in high-stakes fields such as healthcare, space, and cybersecurity. These methods provide explanations in numerical, textual, and/or visual formats. For instance, XAI methods have been adopted in [10] to enhance the transparency of CNN-based models used in spaceborne systems like satellites and aircraft. These systems utilize earth observation technologies to detect maritime vessels for security and emergency response purposes. Leveraging XAI methods help to improve confidence in the AI-enabled recognition of ship forms. In the Cybersecurity domain, [11] demonstrates how XAI methods can enhance the coherency and confidence of AI results generated from DL-based intrusion detection in IoT systems, which is crucial due to the rapidly and evolving digital attacks and large volumes of susceptible big data.

The majority of current XAI DL-based approaches for computer vision tasks provide visual interpretations, highlighting image regions that significantly influence a model's decision. While visual explainability holds great promise in AI-based tools for medical imaging, several usability issues may arise that can hinder their effective adoption in clinical settings [9, 12]. For example, the generated visual explainability can be complex to understand by clinicians, as they may require certain technical knowledge to interpret the output correctly. This creates the need to evaluate the clinical relevance, accessibility and interpretability of these methods to identify their weaknesses and strengths from the human/end user perspective.

In recent years, Grad-CAM (Gradient-weighted Class Activation Mapping) [13], LIME (Local Interpretable Model Agnostic Explanations) [14], and SHAP (SHapley Additive exPlanations) [15] XAI algorithms have gained great popularity in the area of intelligent medical investigation and imaging (e.g. [8, 16, 17]). These methods can be easily integrated with various DL architectures, making them applicable across a broad range of DL models [18]. They mainly explain AI decisions by emphasizing image regions or data features that significantly

contribute to model predictions. This information is valuable for radiologists in second reporting, providing a means to validate and/or assess AI decisions. It can also be incorporated into radiology reports to enhance quality assurance. Moreover, the explainability information can be utilized by clinicians during clinical consultation sessions to inform patients and involve them in the decision-making process.

Despite the valuable interpretability offered by Grad-CAM and LIME algorithms, these techniques have not undergone validation from an end user perspective, utilising real-life clinical scenarios in chest radiology images. Instead, their validation has been primarily relying on data-driven automated evaluation methods (e.g. [17, 19, 20]), which may not always align with the clinical expectations and explainability requirements of medical processes. Evaluating the effectiveness of explanations provided by DL-based diagnostic decisions requires a deep understanding of lung anatomical structures and clinical knowledge of respiratory diseases, aspects that may not be fully captured by computational evaluation methods. Moreover, computational evaluation methods lack the capacity to assess human usability and interaction [21–23], which plays a vital role in gaining trust, acceptance, and eventual adoption of AI-enabled tools in clinical practice. To address this limitation, we propose a user study to evaluate visual XAI systems applied to DL-enabled diagnostic system in chest radiography. This approach is adopted to ensure that the generated explanations are not only accurate and reliable but also comprehensible and useful to end-users. Our contributions can be summarised as follows:

- We develop two clinical case studies showcasing DL-enabled methods for the detection of pneumonia and COVID-19 diseases using two large open access data sets for chest X-rays and CT scans, respectively. In each case study, we examine the performance of two CNN-based models, optimizing their hyperparameters accordingly. Subsequently, we apply the two studied XAI algorithms (Grad-CAM and LIME) to visually explain the classification (diagnostic) outputs.

- We conduct a human centred evaluation for Grad-CAM and LIME XAI systems in chest radiology imaging. To achieve this, we design a qualitative and quantitative user study to evaluate the results obtained from Grad-CAM and LIME tools in explaining the DL diagnostic decisions. Specifically, we evaluate medical practitioner's feedback on XAI results in terms of *comprehensibility*, *confidence*, and *clinical relevance* within the context of chest radiology imaging. The selected evaluation measures has been recently suggested by [24] for evaluating clinical XAI in medical image analysis.

Results obtained from the proposed user study will help us understand the strengths and weaknesses of the two investigated XAI tools from the human (domain expert) perspective, which can inform the development of future iterations of XAI tools in chest radiology imaging. Part of this study is based on the work submitted in [25].

## 2 Literature review

As the use of DL-based methodologies continues to rise, the demand for improving the interpretability of these approaches, particularly in critical domains such as medical image analysis where transparent decision-making holds paramount significance. Hence, explaining the DL results to stakeholders in a coherent and accessible manner is becoming imperative to improve confidence in their outcomes. Several studies have demonstrated the values of XAI methods in assisting clinicians during the diagnostic process using medical images [8, 9]. For example, a study in [26], focusing on tuberculosis diagnosis through chest X-rays, demonstrated that among 13 participating physicians, 77% exhibited enhanced diagnostic accuracy

when utilizing XAI for visual explanations compared to assessments without XAI. According to recent findings in [8, 9], most XAI methods used in medical image analysis apply local post hoc explanations rather than global model-based ones. In simpler terms, these explanations are provided on a case-by-case basis (for each patient) using a neural network that has already been trained, rather than being incorporated during the overall training process on a dataset (for example, for all patients). In response to the need for explainable and reliable AI-enabled COVID-19 diagnostic tools, a study in [27] utilized XAI methods on metagenomic next-generation sequencing (mNGS) samples, employing XGBoost model for the automatic COVID-19 detection and using the combination of LIME and SHAP XAI methods to identify significant COVID-19 biomarkers, enhancing model interpretability and aiding clinicians in understanding the impact of risk factors.

The rapidly growing adoption of XAI algorithms in DL-enabled diagnostic systems has created a great academic and public interest in evaluating AI explainability using various evaluation measures. A recent study in [28] has suggested two main ways for evaluating AI explainability, (a) **objective evaluations**, which involve research studies that utilize objective metrics, including data driven automated approaches to assess explainability methods, and (b) **human-centered evaluations**, which involve human judgment and feedback into the assessment of AI models' explanations. This classification system was also suggested in [29], however, they referred to the two classes as **heuristic-based** and **user-based** metrics.

In the context of medical image analysis, several researchers employed objective evaluation (heuristic-based) methods to evaluate and compare against visual XAI systems. For instance, in [17], Grad-CAM, LIME, and Shapley Additive exPlanation (SHAP) [15]—XAI algorithms were compared and evaluated based on criteria such as consistency, fidelity, sensitivity, and relevance, within the context of a skin lesion image classification task. Another objective evaluation was conducted in [30] by comparing the same three visual XAI methods in the context of COVID-19 detection from CT images. The experimental studies indicated that XAI effectively provides explanations for AI results. However, the latter study lacked using a clear evaluation measure and did not offer distinct findings on the strengths or weaknesses of the compared methods.

Numerous researchers have proposed Human-centered evaluation approaches for evaluating explanations in various classification tasks. In medical diagnosis problems, a research in [21] presented a case study illustrating the application of a human-centered design approach to AI-generated explanations. The case study involved domain analysis, requirements elicitation and assessment, and multi-modal interaction design and evaluation. The study designed explanations for a Clinical Decision Support System (DSS) focused on child health. User studies with experienced paediatricians uncovered necessary explanations during requirements elicitation and assessment. The resulting interaction design patterns were tested in a second user study to evaluate the effectiveness of the explainable DSS in medical diagnosis. The role of explanations on trust and reliance in clinical DSS was also investigated in [31]. A user study was conducted which relieved that clinicians expressed higher satisfaction with DSS outputs when provided with the rationale behind the AI decisions. Another user study was conducted in [32], involving clinicians from two distinct acute care specialties, aiming at identifying specific elements of explainability that could enhance trust in AI models. The study helped to identify the types of explanations that clinicians deemed most relevant and crucial for effective translation into clinical practice.

As far as we are aware, human-centered evaluations for visual XAI methods have not yet been explored in the context of chest radiology imaging research. However, some researchers have delved into clinical evaluation measures specifically designed for assessing visual XAI systems in medical imaging problems. For instance, [24] proposed five clinical guidelines to

support the design and evaluation of clinically-oriented XAI systems. These guidelines encompass Understandability, Clinical Relevance, Truthfulness, Informative Plausibility, and Computational Efficiency. They were developed through a dual perspective, integrating insights from a physician user study on clinical requirements for explanations and the technical expertise of the authors. This study highlights the need for XAI systems to not only provide explanations but to ensure their usability and relevance in supporting informed decision-making in medical practices [21–23]. However, our study assesses both heatmap as well as LIME XAI method, while focusing on chest radiology imaging.

## 3 Clinical case studies

Evaluating the effectiveness of XAI visualizations of chest radiology imaging is most effective when medical professionals are presented with a specific clinical case study in which they can articulate their feedback, needs, and recommendations for improvement [28]. Hence, we engaged with experienced clinicians and radiologists in respiratory disease (from Sandwell And West Birmingham Hospitals NHS Trust) to design and implement two clinical case studies using two open access datasets. This section introduces the dataset and the DL architectures used in our two case studies. We also describe the data preparation process, training protocol, and the results obtained.

### 3.1 Dataset description, pre-processing, and ethics

In this study, we utilize two open-access datasets:

**1. Dataset 1: Chest X-ray Images for pneumonia detection**. This dataset contains 5,863 X-ray images (JPEG) from Guangzhou Women and Children's Medical Center, split across two categories—pneumonia/normal [33]. The chest X-ray images (anterior-posterior) were retrospectively selected from cohorts of paediatric patients (aged one to five years) from Guangzhou Women and Children's Medical Center. Imaging was performed as part of the patients' routine medical care. The original authors obtained Institutional Review Board (IRB) approvals and collected the data in compliance with the United States Health Insurance Portability and Accountability Act (HIPAA) [34] and the Declaration of Helsinki.

**2. Dataset 2: CT Scans for SARS-CoV-2 (COVID-19) detection**: This dataset comprises 2,482 CT scans from patients in Sao Paulo, with 1,252 CT scans positive for COVID-19 infection and 1,230 CT scans are negative [35]. Data providers ensured the necessary IRB approvals and obtained all required patient consents.

The two datasets had been pre-processed and anonymized by the data providers. For example, the chest X-rays were screened to remove low quality and unreadable scans, the remaining images were then annotated by chest radiologists for diagnosis labelling and all metadata and clinical image headers were removed to erase patient data. Both datasets were cropped to optimal size for use in machine learning models.

Both datasets are ethically approved for research and are licensed under the Creative Commons Attribution 4.0 International license, which allows for sharing, copying, and modifying the datasets for research purposes. Prior to utilizing these datasets and commencing our user study, ethical approval was obtained from Aston University (Ethics Approval Number: 234700–06). Participants received a detailed email outlining the research project objectives and their role in the study. Consent was obtained before they accessed the online survey, which was open for nearly a month to facilitate participation.

## 3.2 Deep learning- based diagnostic systems

The proposed DL-based Diagnostic systems consist of two main components: Convolutional Neural Network (CNN)-based models for image classification, and XAI-based explanation generations framework. CNN-based models were employed for the automatic diagnosis of chest diseases from X-ray and CT scans due to their effectiveness in capturing spatial hierarchies and patterns within medical imaging data, enabling robust feature extraction for accurate diagnostic predictions. To this end, we trained several CNN models, including VGG-19, ResNet-50, ResNet-101, DenseNet169 and MobileNetV2 on both datasets. These architectures have shown promising results in similar automatic diagnostic tasks in medical imaging [4, 5]. Additionally, previous work has shown that effective image classification performance can be achieved through pre-trained models fine-tuned on specific tasks [6, 16].

**3.2.1 Experimental settings.** Experiments were conducted using the online Jupyter Notebook within the Anaconda Navigator in the Python programming language, employing TensorFlow frameworks. The hardware specifications include an AMD Ryzen 7 PRO 5850U Processor with Radeon Graphics (1.90 GHz), 16 GB RAM, a 64-bit operating system, and an x64-based processor. The training process involved the use of the following libraries: TensorFlow, Keras, NumPy, Pandas, and Matplotlib.

The evaluation measures used to measure their performance are accuracy, precision, recall, F-measure (F1 score). These measures are based on true positive (TP), true negative (TN), false positive (FP), and false negative (FN). These measures are described in the equations below.

**Accuracy**: measures the overall correctness of the model. $Accuracy = \frac{TP+TN}{TP+TN+FP+FN}$

**Precision**: (also called positive predictive value) measures the accuracy of positive predictions. $Precision = \frac{TP}{TP+FP}$

**Recall**: (also called sensitivity or true positive rate) measures the ability of the model to capture all the relevant instances. $Recall = \frac{TP}{TP+FN}$

**F1 Score**: the harmonic mean of precision and recall, providing a balance between the two metrics. $F1 = \frac{2 \times Precision \times Recall}{Precision + Recall}$

We implemented several techniques throughout our study to monitor and mitigate overfitting. This includes applying regularization techniques, specifically L2 regularization and dropout to penalize model complexity and minimizing the risk of overfitting. For instance, a dropout rate of $1^{-0.5}$ was used in the MobileNetV2 and ResNet-101 models (best performing) to classify chest X-ray images and CT scans into pneumonia and normal, and COVID-19 and Non-COVID-19 cases, respectively. We also utilised early stopping in the models to stop training the model after its optimal number of iterations has been reached. Furthermore, both the training and validation loss curves were continuously monitored to ensure that no significant divergence between these curves occurred, which is often a good indicator of overfitting.

**3.2.2 Results.** The results obtained for Dataset 1, reported in Table 1, reveal that MobileNetV2 model outperformed other compared algorithms in diagnosing pneumonia in chest X-rays. While the results obtained for Dataset 2, reported in Table 2, show that ResNet-101

**Table 1. Performance metrics (on testset) of deep learning models on Dataset 1 (Chest X-ray images for pneumonia detection).**

| Model | Accuracy | Precision | Recall | F1-Score |
|---|---|---|---|---|
| MobileNet V2 | 91.03% | 88.13% | 98.97% | 93.24% |
| VGG19 | 75.16% | 80.54% | 55.60% | 65.78% |
| ResNet 101 | 88.93% | 75.79% | 82.43% | 79.96% |
| DenseNet 169 | 90.06% | 81.46% | 66.06% | 72.96% |

**Table 2. Performance metrics (on testset) of deep learning models on Dataset 2 (CT Scans for COVID-19 detection).**

| Model | Accuracy | Precision | Recall | F1-Score |
|---|---|---|---|---|
| ResNet 101 | 98.39% | 98% | 98% | 98% |
| ResNet 50 | 93.58% | 91.28% | 96.22% | 93.68% |
| MobileNet V2 | 93.30% | 100% | 96.83% | 98.39% |
| VGG19 | 50.40% | 50.60% | 50% | 67.20% |

model performed best in COVID-19 detection using chest CT scans. In all experiments, Slice Size was fixed to 224 x 224 and ReduceLROnPlateau was used for the Scheduler.

Figs 1 and 2 show the training and validation loss and accuracy plots for the best performing models, in Dataset 1 and 2, respectively. The loss plot for Dataset 1 (Pneumonia) shows steady decrease while the accuracy plot shows steady increase over time before early stopping terminates the training to avoid overfitting. Similarly the loss plot for Dataset 2 (COVID-19) shows steady decrease while the accuracy plot shows steady increase over time, however all epochs run until completion. Fig 3a shows the Receiver Operating Characteristic (ROC) plot with Area under the ROC Curve(AUC) score for the best performing model for Dataset 1 which presents an AUC score of 0.88 in classifying Chest X-rays with and without Pneumonia, while Fig 3b shows the ROC plot with AUC score for the best performing model for Dataset 2 which presents an AUC score of 1.0 in classifying CT scans with and without COVID-19.

## 3.3 Visual explainability models

As indicated in section 1, this paper aims to evaluate XAI methods in AI-enabled chest radiology diagnostic systems. Based on our initial experimental findings, Grad-CAM [13] and LIME [14] provide more stable and accurate localized explanations compared to SHAP [15] in both clinical case studies (image classification tasks). Therefore, in this paper, we selected LIME and Grad-CAM methods due to their superior performance in delivering accurate, relevant, and

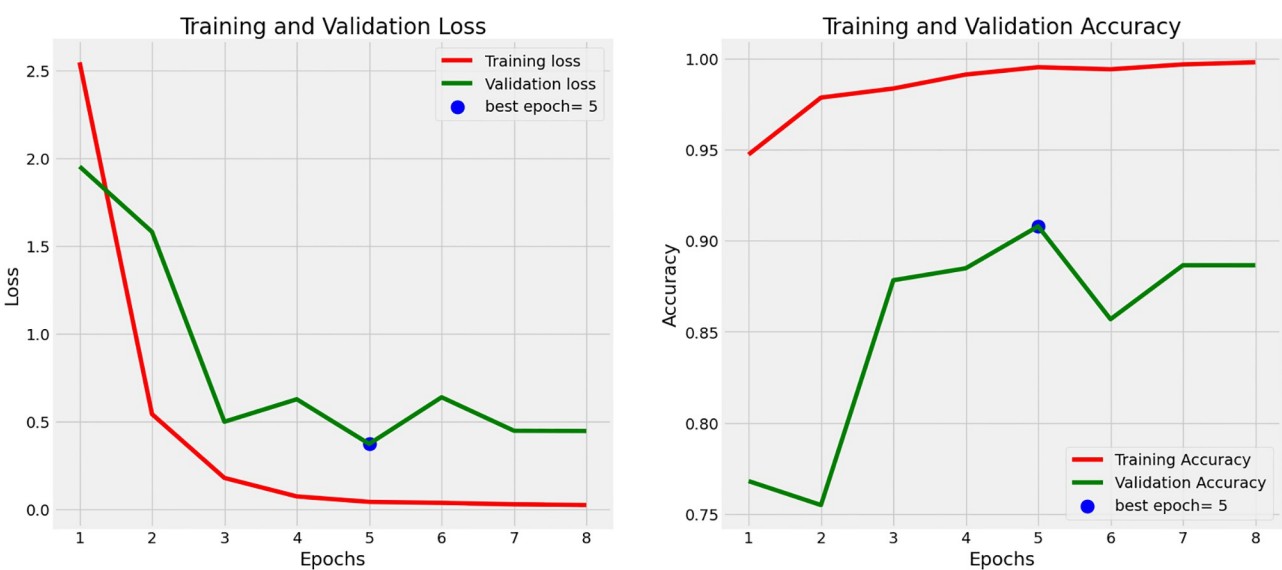

**Fig 1. Accuracy and loss for the deep learning models used for Dataset 1 (Chest X-ray images for pneumonia detection).** The plots illustrate the model's performance and convergence during the training process.

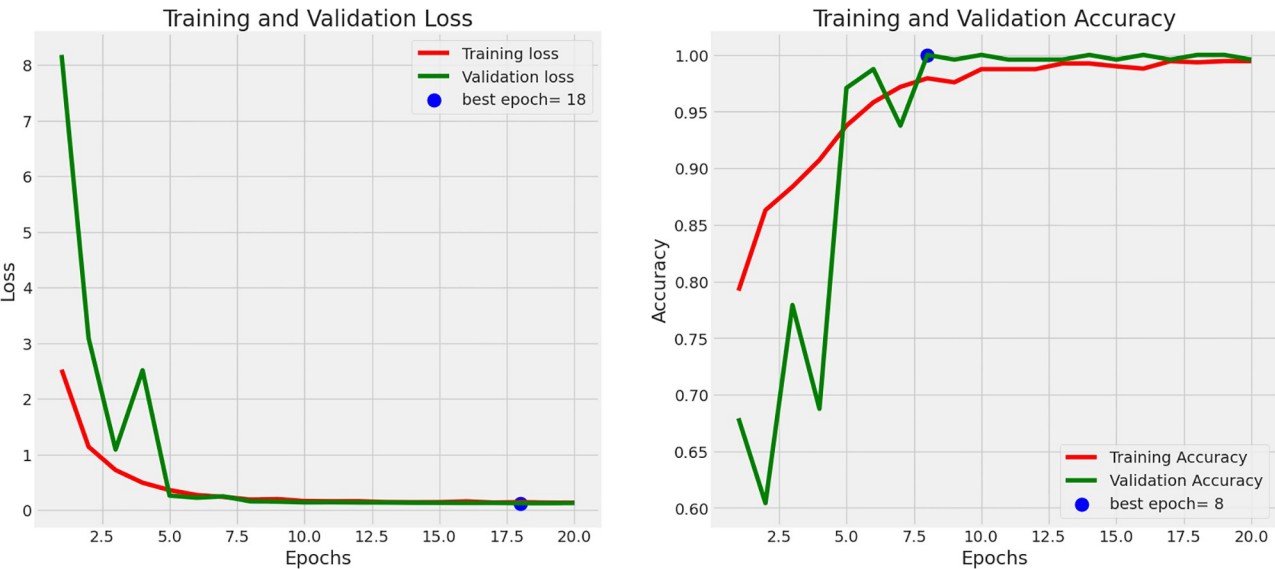

**Fig 2. Accuracy and loss for the deep learning model used for Dataset 2 (CT Scans for COVID-19 detection).** The plots illustrate the model's performance and convergence during the training process. (**a**) ROC curve for Dataset 1 (Chest X-ray Scans for Pneumonia detection) (**b**) ROC curve for Dataset 2 (Chest CT Scans for COVID-19 detection).

stable explainability results. Evidence from recent literature supports this choice. For instance, a study in remote sensing image classification [36] compared the performance of ten different XAI methods and found that Grad-CAM and LIME were the most interpretable and reliable. Similarly, a research in [17] comparing Grad-CAM, SHAP, and LIME in the context of

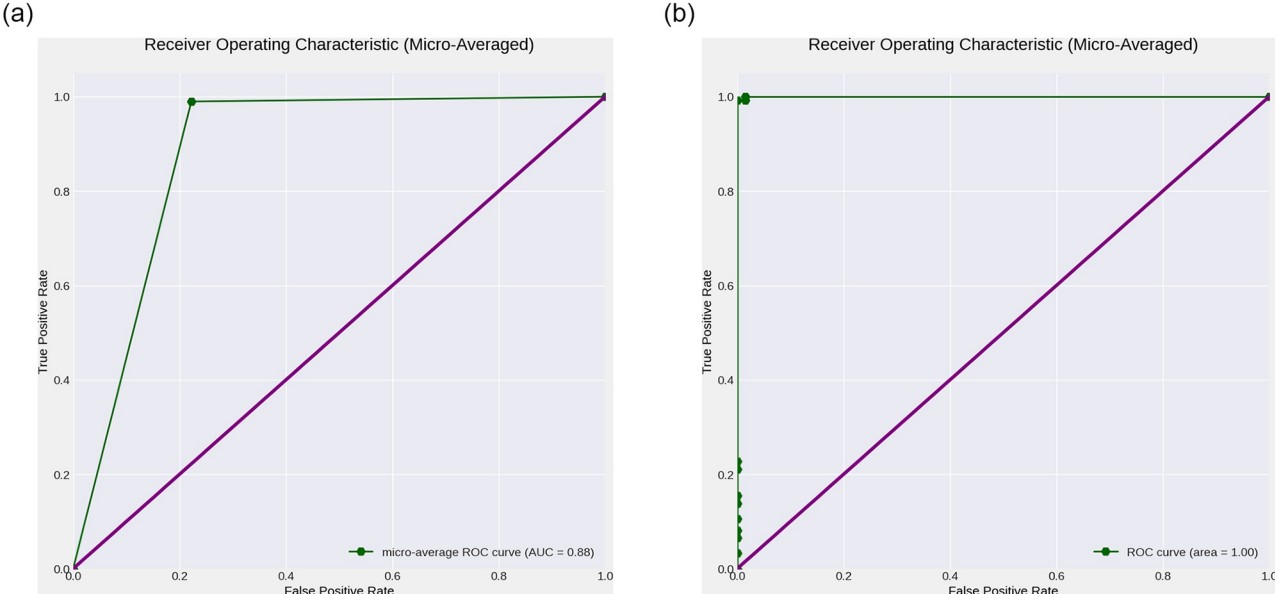

**Fig 3. Receiver Operating Characteristic (ROC) (green) with Area under the ROC Curve (AUC) generated for the clinical case studies (Datasets 1 and 2).** ROC and AUC plots are generated for the best performing deep learning models for classifying Chest X-rays with and without Pneumonia, and CT scans with and without COVID-19. (a) ROC curve for Dataset 1 (Chest X-ray Scans for Pneumonia detection), (b) ROC curve for Dataset 2 (Chest CT Scans for COVID-19 detection).

medical imaging concluded that Grad-CAM and LIME were more reliable, whereas SHAP was not the best for local accuracy in this application. This is consistent with another findings in [15], which highlights that while SHAP provides comprehensive feature importance in non-imaging datasets, it may produce less stable explanations in complex image classification tasks, leading to potential inconsistencies. These findings underscore the reliability and relevance of LIME and Grad-CAM in our study, facilitating better insights and trust in the model outputs.

**3.3.1 Gradient Class Activation Mapping (Grad-CAM).** Class Activation Maps (CAMs) algorithm [37] works by highlighting significant areas or gradients within an image, aiding CNNs in predicting the image's class. CAM generates weighted activation maps in the last convolutional layer of the CNN, determining neuron contributions to predictions. The average of these weighted connections is used to construct feature maps. For CAM to function correctly, the CNN must conclude with a single fully connected layer connected to a global average pooling layer, and the CAM must match the image dimensions through bilinear interpolation.

Grad-CAM [13] is the extended version of CAM and it enhances interpretability by computing the gradients of the predicted class score with respect to the feature maps of the last convolutional layer, yielding more detailed and accurate visual explanation. The class-discriminative localization map (L) for a specific class c, can be calculated as follows:

$$L_c = \sum_k \alpha_k^c A_k$$

Where $k$ denotes the index of the last convolutional layer, $\alpha_k^c$ is the weight associated with the k-th location for class c, and $A_k$ is the activation of the k-th location in the last convolutional layer.

In this study, Grad-CAM was applied to last layer of the best performing (MobileNetV2's) deep learning models for predicting pneumonia and COVID-19 from chest X-ray and CT images. The heatmaps generated was resized to match the original image and overlaid to produce the final Grad-CAM image output.

**3.3.2 Local Interpretable Model-Agnostic Explanations (LIME).** LIME [14] works by initially segmenting a given image into superpixels, homogeneous groups of pixels that simplify image representation, and then approximating the complex model's behavior locally by fitting an interpretable model to perturbed instances around the original input. The mathematical representation involves constructing a locally weighted linear model $g_z(x)$ that approximates the model $f$ in the neighborhood of a given instance $x$:

$$g_z(x) = \arg\min_g L(f, g, \pi_x) + \Omega(g_z)$$

where $L$ is a loss function measuring the difference between $f(x)$ and $g_z(x)$, $\pi_x$ is a proximity measure, and $\Omega(g_z)$ is a regularization term.

In the context of this study, the LIME image explainer model was applied to the results of the ResNet-101 deep learning diagnosis of COVID-19 from chest CT scans. The LIME model not only highlighted the most activated weighted neurons used in the diagnosis but was also configured to mask areas with minimal neuron contributions.

# 4 XAI human-centered evaluation method

Due to the popularity of the LIME and Grad-CAM methods, we conducted a user study to evaluate their usefulness in the context of chest radiology imaging.

### 4.1 Participants and recruitment

To be eligible to participate in our study, individuals needed to be clinicians or radiologists, with relevant medical speciality and good understanding in medical imaging as well as diagnostic decision-making. Online survey was deemed the most appropriate approach to disseminate the AI explainability results, allowing participants to access and respond to the survey at their convenience. To facilitate our recruitment process, we leveraged our professional network and engaged with National Health System (NHS) in UK. This organization assisted in promoting the study through their websites and mailing lists.

### 4.2 User study ethical statement

Before commencing the user study, ethical approval was obtained from Aston University (Ethics Approval Number—234700–06). Participants received an initial detailed email communication outlining the research project objectives and their role. After receiving positive responses, participants were provided with a link to the online survey, accompanied by a participant information sheet explaining their involvement, data handling procedures, and survey result utilization. Online consent (collected through online Microsoft form) was obtained before participants proceeded to complete the online survey questionnaire. No minors (children) were included in this project. The survey remained open for almost one month (22nd of August 2023—20th of Sep 2023) to facilitate participation. The need for consent was waived by Aston University ethics committee.

### 4.3 User study design

While there are different ways to conduct user studies (interviews or focus groups), online questionnaire was deemed as the most suited method for this study, as it allowed us to organize and compare responses in a systematic way. It also allowed analysing feedback results in both quantitative and qualitative ways. To ensure comprehensive feedback, we employed a combination of open and closed questions, allowing participants to express their information preferences effectively and to elaborate on certain aspects. Consulting domain experts before conducting a survey is essential for ensuring the quality, relevance, and effectiveness of the questionnaire. To this end, before designing our user study we interviewed radiology and clinical specialists to aid in designing relevant survey questions.

Our user study aimed to achieve four primary objectives:

1. Understand medical practitioners' awareness and sentiments towards the use of AI and XAI in medical imaging practices. This understanding was crucial for identifying potential biases in survey responses related to the studied XAI methods.

2. Evaluate the effectiveness of Grad-CAM and LIME visualization methods in the context of the proposed case study—automatic diagnosis of pneumonia and COVID-19 from X-ray and CT scans, respectively. The human-centered evaluation have been conducted in terms of *usefulness*, *understandability*, and *trust* of the explainability, in line with the recent recommendations in [24].

3. Compare between Grad-CAM and LIME XAI methods, from the medical domain experts' perspective in the context of chest radiology diagnostic tasks.

4. Solicit feedback and recommendations from medical domain experts on how to enhance the presentation, usability, and explanation content of XAI tools in the context of chest radiology imaging practices.

*I.    What is your medical speciality? Options - [Radiologist / Orthopaedist / ICU Doctor or Other medical speciality]*

*II.    (If 'Other'), Please type your speciality in the text field below.*

*III.    What is the level of your experience in the medical field? Options - [Less than 1 year (e.g., medical student) / 1-3 years / 3-5 years / 5-10 years / More than 10 years]*

*IV.    How many years of experience you have in analysing radiology images? Options - [Less than 1 year (e.g., medical student) / 1-3 years / 3-5 years / 5-10 years / More than 10 years]*

**Fig 4. Constructed questions for questionnaire—part 1.1.** Questions aim to collect basic information concerning participants' medical speciality, medical imaging knowledge, and overall experience in the medical field.

To achieve the above objectives, we structured the survey into four distinct sections as follows:

**4.3.1 Part 1: Medical practitioners' characteristics, attitude and sentiment towards AI and XAI in medical imaging.**   The first part seeks to elicit basic information concerning participants' medical speciality, medical imaging knowledge, and overall experience in the medical field. This part also aimed at capturing participants' current awareness of AI technology in medical imaging, and their acceptance, trust, and sentiment towards AI-support of medical diagnosis. We also explored participants' familiarity with the concept of XAI in medical imaging. Questions utilised a scale of 1 to 5, with 1 being the lowest score and 5 being the highest, which are commonly used in surveys to measure respondents' attitudes, opinions, or experiences on a particular topic. The scale allows participants to indicate their level of agreement or disagreement with a statement or the extent to which they endorse a particular sentiment. Figs 4 and 5 show some of the questions constructed in this part. (The complete list of questions is provided as a supplementary material to this paper).

**4.3.2 Part 2—Presentation of visual explainability results.**   The second part of the survey presents the XAI results obtained from our two clinical use cases (explained above) illustrating the application of XAI in the DL diagnosis of pneumonia and COVID-19 from chest

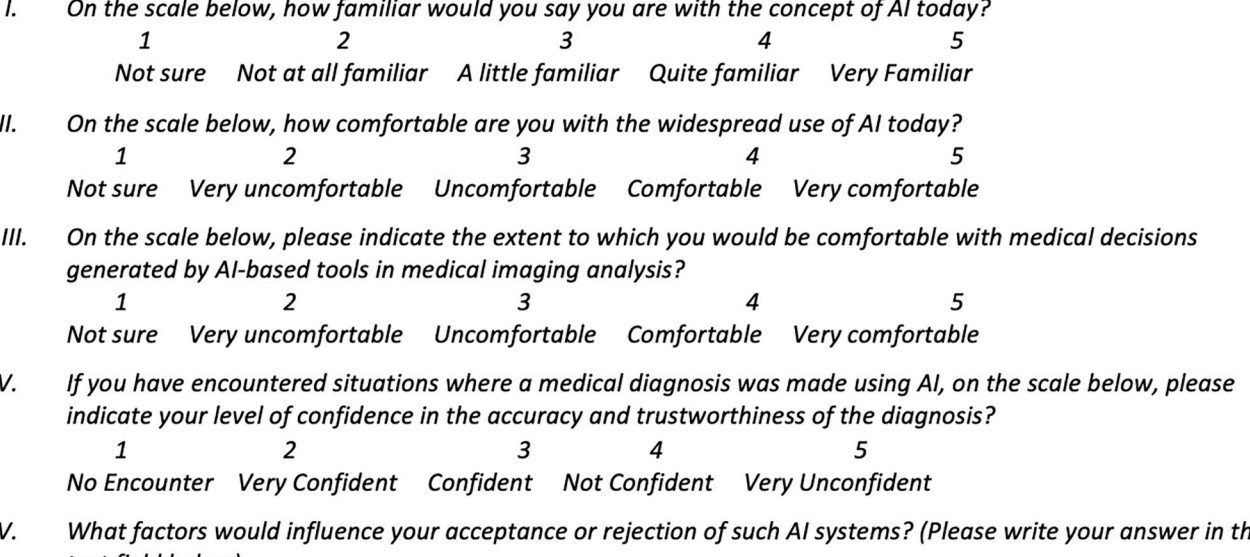

**Fig 5. Constructed questions for questionnaire—part 1.2.** Questions aim to explore participants' familiarity with the concept of XAI in medical imaging.

X-ray and CT scans, respectively. We provided an introduction including a general description of each clinical use case, including a brief description of the data, techniques and its performance. Consequently, we provide a set of four explanatory case results, each for pneumonia and COVID-19, generated through DL diagnosis using Grad-CAM and LIME. Guidance statements alongside the displayed XAI results images were presented to brief participants about the dataset specifics and the images that will be showcased. Figs 6 and 7 show examples of similar data presented in the survey. In Fig 6, images (a) and (e) depict original chest X-ray scans of patients diagnosed with (positive) and without (negative) pneumonia, respectively. Images (b) and (f) show the Grad-CAM XAI Results. The overlayed heatmaps highlight the important regions of interest (ROIs) that contributed most to the model's decision. Images (c) and (g) show the superpixel regions of the image. Images (d) and (h) show the LIME XAI Results. The images display the significant ROIs identified by the AI model, with less significant regions obscured by grey. In Fig 7, images (a) and (e) depict original chest CT scans of patients diagnosed with (positive) and without (negative) COVID-19, respectively. Images (b) and (f) show the Grad-CAM XAI Results. The overlayed heatmaps highlight the important regions of interest (ROIs) that contributed most to the model's decision. Images (c) and (g) show the superpixel regions of the image. Images (d) and (h) show the LIME XAI Results. The images display the significant ROIs identified by the AI model, with less significant regions obscured by grey.

**4.3.3 Part 3: Evaluation and comparison of XAI tools.** The goal of this section is to assess the quality of the explanation provided by both Grad-CAM and LIME, and assess their effectiveness in influencing clinical decision-making within radiology workflow. Drawing from the research in [24], we've identified three main evaluation criteria for evaluating visual XAI in medical imaging.

- **Clinical Relevance**: assess whether explainability results are *useful*, *usable* and more importantly contain *accurate* information for medical decision-making.

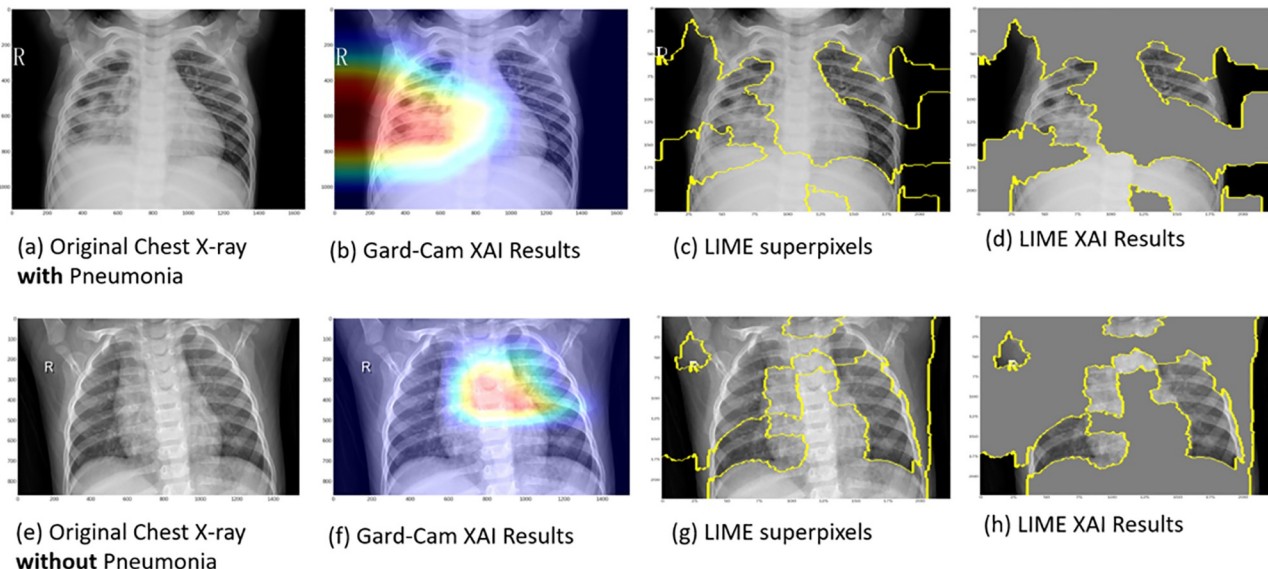

**Fig 6. Explainable AI (XAI) visualization results for clinical case study one.** This figure illustrates XAI techniques (Grad-CAM (b and f) and LIME (d and h)) applied to chest X-ray images for pneumonia detection, highlighting the regions and features of the images that the deep learning model focuses on to make its predictions.

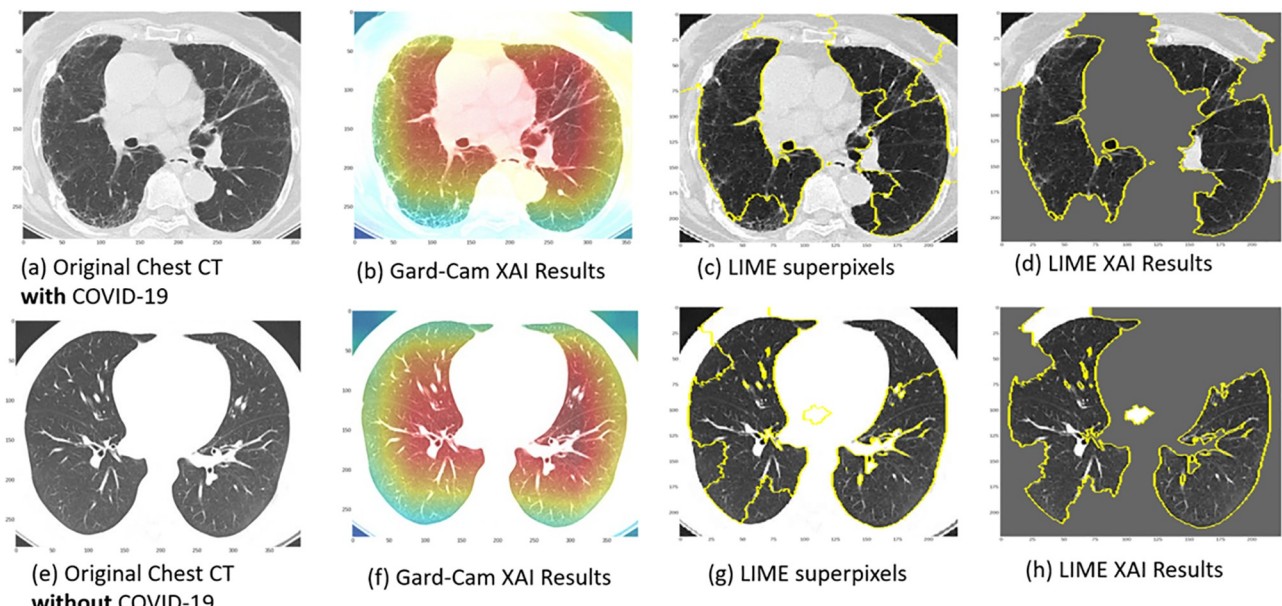

**Fig 7. Explainable AI (XAI) visualization results for clinical case study two.** This figure illustrates XAI techniques (Grad-CAM (b and f) and LIME (d and h)) applied to chest CT images for COVID-19 detection, highlighting the regions and features of the images that the deep learning model focuses on to make its predictions.

- **Comprehensibility**: evaluate the *coherency* of the explainability results, and whether the presentation format is easily understood by clinicians without requiring technical expertise.

- **Confidence/Trust**: evaluate the extent to which explanations truthfully reflect the AI model's decision process, influencing trust and confidence in the results.

The above criteria emerged through collaborative effort and expertise from both clinicians and medical image analysis experts. Using the questionnaire, we aim to quantify how well each XAI tool (Grad-CAM, LIME) aligns with each of the above evaluation metrics. Figs 8 and 9 show some of the questions constructed in this part.

**4.3.4 Part 4: Future recommendations.** This section is dedicated to soliciting suggestions and feedback for improving the explainability of AI models in medical imaging from the users' standpoint. Specifically, we aim to gain insights into the limitations of current XAI visualizations and solicit recommendations for enhancing the design and development of XAI in medical imaging. Additionally, we explore participants' perspectives on Multi-modal explainability in medical imaging, which involves integrating both textual and visual explanations, and its impact on improving the coherency and trust of AI models.

The concept of human-centered explainability has recently received significant attention in the literature and among medical imaging researchers [21–23]. Consequently, we seek participants' insights on the inclusive design of AI and XAI systems, exploring the roles clinicians or radiologists can play in the design and development process. Lastly, we gauge participants' agreement or disagreement regarding the potential of XAI visualizations to enhance radiology practices overall. Questions in this section have been mainly presented in an open style to allow participants the opportunity to elaborate on these crucial inquiries. Fig 10 shows some of the questions presented in this part.

I. *On the scale of 1 to 5 below (Where 1 is the lowest rating and 5 is the highest rating), how would you rate the usefulness of the **XAI tool 1: Grad-CAM (Heatmaps)** visual explanations presented above regarding the clinical impact on your clinical decision-making?*

II. *On the scale of 1 to 5 below (Where 1 is the lowest rating and 5 is the highest rating), how would you rate the coherency or comprehensibility of the **XAI tool 2: LIME (Greyscale with yellow outlines)** visual explanations presented above?*

III. *Are there any specific aspects of the visual explanations presented above that you find particularly challenging or confusing to understand? (Please write your answer in the text field below) – [Text field]*

IV. *Between XAI tool 1: Grad-CAM (Heatmap) or the XAI tool 2: LIME (Outlines), which visualization method did you find more understandable and useful?*

V. *On the scale below, please indicate your level of confidence in the accuracy and trustworthiness of the visualizations presented.*

| *1* | *2* | *3* | *4* | *5* |
|---|---|---|---|---|
| *No Encounter* | *Very Confident* | *Confident* | *Not Confident* | *Very Unconfident* |

VI. *Did the explanations provided by the tool influence your trust in the AI system in the analysis of radiology imaging?*

**Fig 8. Constructed questions for questionnaire—part 3.1.** Questions aim to assess the quality of the explanation provided by both Grad-CAM and LIME, and assess their effectiveness in influencing clinical decision-making within radiology workflow.

VII. *From the visualizations presented for **XAI tool 1: Grad-CAM (Heatmaps)**, do you agree or disagree that the readability of the visualization results is negatively impacted by the colouring scheme used?*

*[Strongly Agree / Agree / Neither agree nor disagree / Disagree / Strongly Disagree]*

**Fig 9. Constructed questions for questionnaire—part 3.2.** Questions aim to assess the impact of the coloring scheme on the XAI visual results.

I. *Considering the visual explanations presented above, what improvements or enhancements would you suggest for making the visualizations more effective and easily understandable for medical professionals / radiologists? - [Text field]*

II. *Do you think that using textual explanations (image descriptive sentences) along with the visualizations will improve coherency and clearance of the explanations provided by the Explainable AI tool? - [Not sure / Yes / No]*

III. *Do you agree or disagree that clinicians or radiologists should play a role in the design and development of XAI visualization systems for medical imaging?*
*[Strongly Agree / Agree / Neither agree nor disagree / Disagree / Strongly Disagree]*

IV. *Is there more feedback or recommendations you would like to share about the presented XAI visualizations in radiology imaging analysis?*

**Fig 10. Constructed questions for questionnaire—part 4.1.** Questions aim to collect recommendations for improving the explainability of AI models in medical imaging from the users' perspective.

Following the completion of the questionnaire, a pilot study involving an experienced radiologist ($>$ 15 years) was conducted to assess its appropriateness for clinicians. The pilot study focused on validating the questions in alignment with the study's medical objectives, evaluating the coherence of the question flow, and optimizing the use of medical terminology for clearer communication within the medical field. Following the pilot study, adjustments were implemented in specific sections of the questionnaire based on feedback received from the experienced radiologist.

## 5 User study results

### 5.1 Part 1 results: Clinicians characteristics, attitude and sentiment towards AI and XAI in medical imaging

**5.1.1 Participant characteristics.** A total of 26 clinicians participated in the survey, with 8 identifying as Radiologists, and the remaining 18 having various medical specialties (see Figs 11 and 12). While most participants possess extensive experience in their medical fields and radiology image analysis, less than 40% have experience in the relatively new domain of AI-based medical imaging. Specifically, out of the 26 participants, 16 reported having no prior exposure to AI-based medical imaging tools in their practice (see Fig 13) below for a visual representation of the distribution of participants' medical experience in medical imaging and with the use of AI-based medical imaging tools.

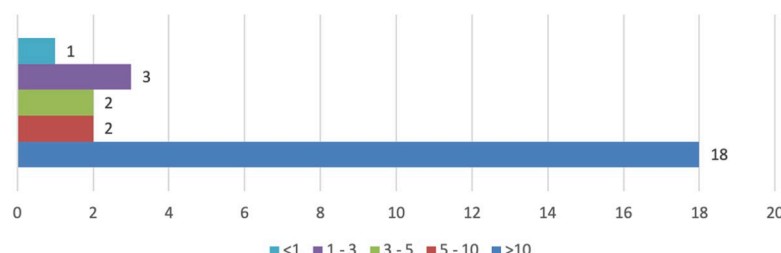

**Fig 11. Distribution of participant's total medical experience.** The figure indicates that 18 participants have more than 10 years of experience, showcasing the overall experience levels within the participant group.

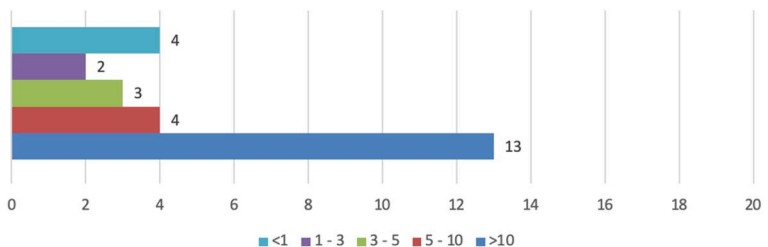

**Fig 12. Distribution of participants' experience analyzing radiology images.** The histogram indicates that 13 participants have more than 10 years of experience, highlighting the expertise level within the group.

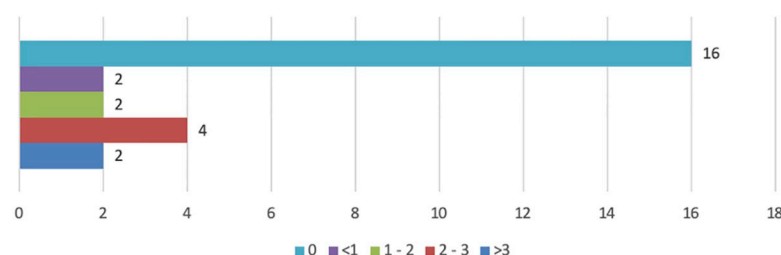

**Fig 13. Distribution of participants' experience with AI-based medical imaging tools.** The figure reveals that 16 participants have zero experience with AI-based medical imaging tools, highlighting a significant portion of the group with no prior exposure to this technology.

**5.1.2 Attitudes towards AI in medical imaging.** Figs 14 and 15 illustrate participants' familiarity with the AI concept and their comfort levels with the general widespread use of AI. With regards to the comfort in using AI-based tools in medical diagnosis, opinions almost split between being Not sure and Comfortable (see Fig 16). On the other hand, most participants, fourteen in total, reported poor confidence in AI-based diagnostic decisions (see Fig 17).

Participants shared factors that would enhance their acceptance of AI-based diagnostic systems. The responses primarily emphasised the importance of human supervision of the diagnostic system, the need for highly accurate results, transparency in the training and testing processes, clear understanding of the ethical aspects of AI-based diagnosis, transparency in the decision-making process, and the enhancement of their own AI knowledge. Example answers are provided below, "*The size and source of the database*", "*I need to understand the way they analyse the data*", and "*Radiologists monitoring and supervision*".

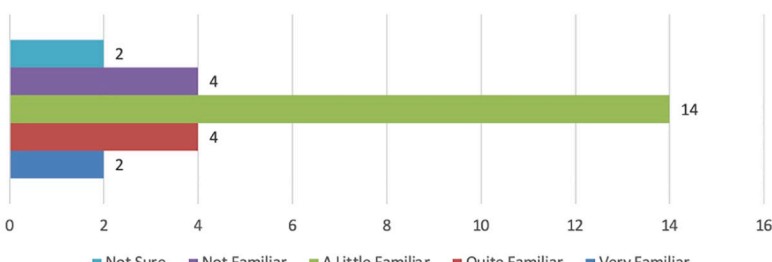

**Fig 14. Distribution of participants' familiarity with AI.** The figure shows that 14 participants reported being "little familiar" with AI, highlighting the varying levels of AI knowledge among the participants.

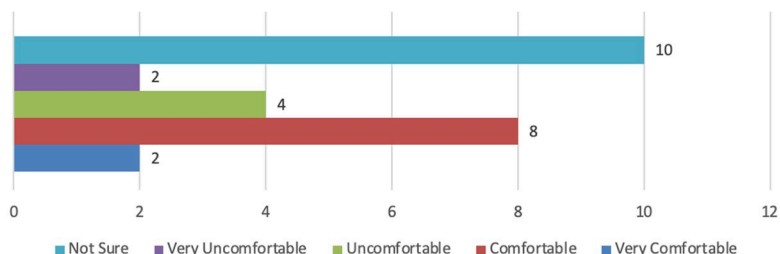

**Fig 15. Distribution of participants' comfort with the general widespread use of AI.** The figure shows that most participants are feeling very comfortable with the general widespread use of AI.

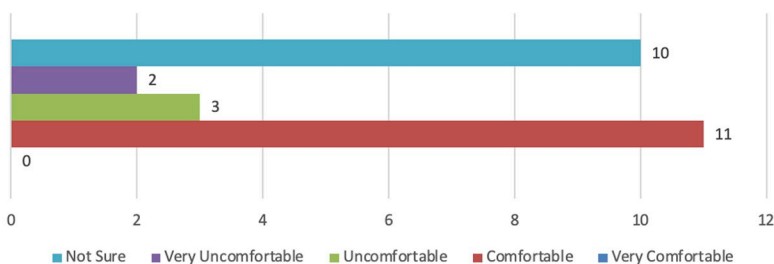

**Fig 16. Distribution of participants' comfort with the medical decisions generated from AI-based tolls.** The figure shows that opinions almost split between being Not sure and Comfortable.

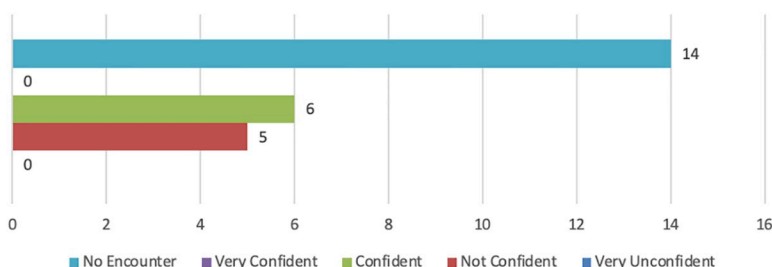

**Fig 17. Distribution of participants' confidence in AI-based diagnostic tools.** The figure shows that most participants, fourteen in total, reported poor confidence in AI-based diagnostic decisions.

Furthermore, Fig 18 illustrates that nineteen participants consider it crucial for medical practitioner to understand the rational of the AI decision in medical imaging systems, while only five participants viewed this aspect as unimportant.

**5.1.3 Awareness of XAI in medical imaging.** Fig 19 displays the responses of participants when questioned about their awareness of XAI tools in medical imaging. Only four participants had prior knowledge of XAI tools, and among them, two participants expressed a belief in the effectiveness of these tools in explaining results in medical imaging tasks (see Fig 20).

**In the first part of our findings**, it's evident that clinicians hold a range of sentiments toward AI tools, particular in its use for diagnostic purposes, varying from 'neutral' to 'positive.' While they recognize the progress AI has made in healthcare, most haven't had extensive exposure to the ins and outs of these technologies—both in theory and practice. This highlights a shared eagerness among clinicians to deepen their understanding of these advancements, with the goal of building greater confidence in applying them within the medical field. Furthermore, the lack of awareness of XAI among medical experts was also highlighted in the

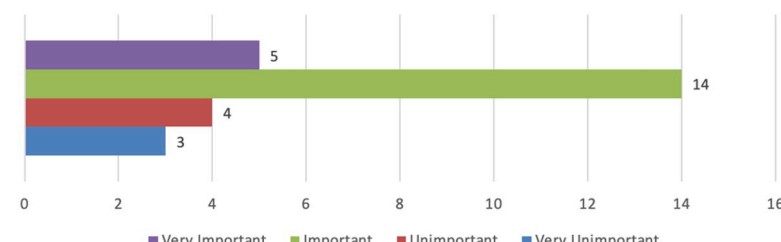

**Fig 18. Distribution of participants' support for understanding the decision-making process of AI algorithms used in medical imaging.** The figure illustrates that nineteen participants consider it crucial for medical practitioners to understand the rationale of AI decisions in medical imaging systems, while only five participants view this aspect as unimportant.

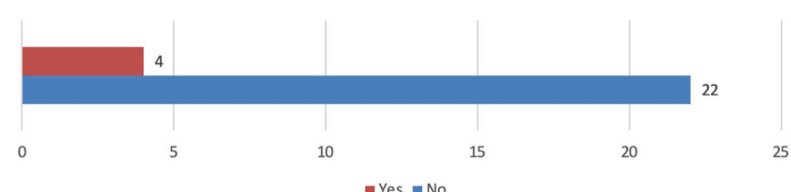

**Fig 19. Distribution of participants' awareness of XAI.** The figure shows that most participants reported being poor familiarity of XAI in medical imaging.

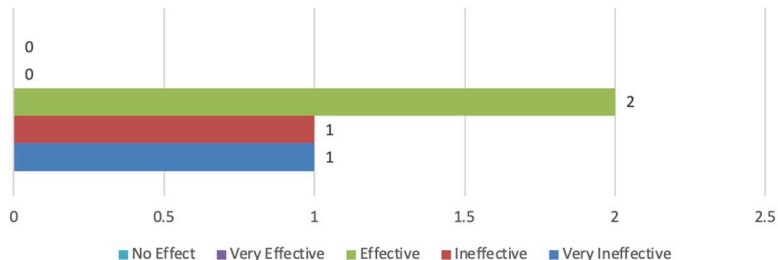

Effectiveness of Explainable Artificial Intelligence (XAI) tools in providing insight and explanations for AI-generated results in medical imaging analysis

**Fig 20. Distribution of participants' belief in the effectiveness of XAI tools insights.** The figure shows that most participants didn't respond to this question due to their poor familiarity with the XAI concept.

results. This is mainly due to the fast-paced technological advancements in AI technology, which makes it hard for clinicians to keep up with the latest and complex developments. Also, it is worth nothing that XAI applications are not yet widely implemented in clinical settings nor in medical education, hence, clinicians have limited exposure to these technologies in their daily routines.

## 5.2 Part 3 Result: Evaluation and comparison of XAI tools

As outlined above, a primary objective of this research is to evaluate and compare Grad-CAM and LIME in explaining AI decisions for diagnostic chest diseases in radiology images, using the two presented case studies. The evaluation and comparison will encompass three key aspects: (a) **clinical relevance**, encompassing *usefulness*, *usability*, and *accuracy*; (b) **comprehensibility**, gauging the coherency of the explainability results; and (c) **confidence** in AI results. Questions were presented on a scale from one to five, with **one** representing the **lowest** and **five** the **highest** rating.

**5.2.1 Clinical relevance.** In terms of the **Grad-CAM** usefulness, twenty participants expressed positive evaluations, providing scores of three or higher on a five-point scale (see Fig 21). These participants found the heatmaps beneficial for focusing on critical areas, confirming medical suspicions, and improving precision. Notably, chest X-ray visualizations were deemed to offer clearer explanations compared to chest CT scans. Participants offering high scores provided diverse justifications, such as "*Because XAI improve precision in making diagnosis*", "*Easy to screen for pneumonia.*", and "*clear and specific.*". On the other hand, six participants rated the heatmap visualizations below three for usefulness, expressing concerns about potential cognitive bias and inaccurate explanation. Participants assigning lower scores provided varied justifications, including "*Introduces cognitive bias.*", "*Loss of focus on original image*", and "*The AI produced images are quite confusing to me as I am not familiar with.*"

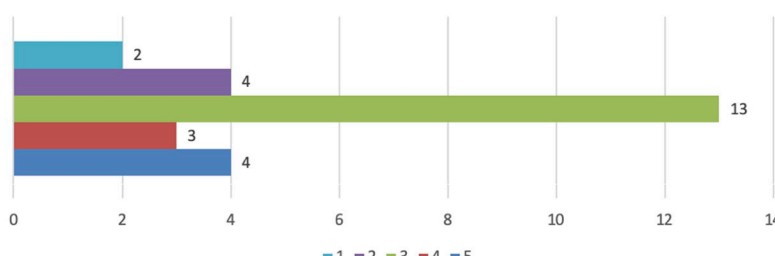

Grad-CAM (Usefulness)

**Fig 21. Grad-CAM clinical relevance (Usefulness).** The figure shows that most participants expressed positive evaluations on the usefulness of the Grad-CAM method in explaining the AI results.

When querying participants about the effectiveness of the coloring scheme in Grad-CAM visualizations, thirteen indicated that the colored heatmaps had a negative impact on scan readability (see Fig 22). Ten participants expressed no clear opinion, and only three disagreed. The primary concern raised was that the color scheme posed readability challenges, particularly for users with color blindness.

Regarding the usefulness of the LIME visualizations, unlike Grad-CAM, only 2 participants assigned a score of 5 for the usefulness criteria (see Fig 23). Furthermore, while 6 participants rated Grad-CAM below three, 9 participants scored LIME less than 2 for the usefulness criteria. They expressed concerns about misclassification of specific regions of interest and apprehension that the greyed-out areas in the visualizations might lead to the omission of essential details in the input image. One participant succinctly explained, "*I don't find it very useful.*"

**5.2.2 Comprehensibility.** When assessing the comprehensibility/coherency of Grad-CAM visualisations, twenty-two participants rated the heatmap visualisations positively, with scores of three or higher (see Fig 24). Some found the visualisations to have ample detail in

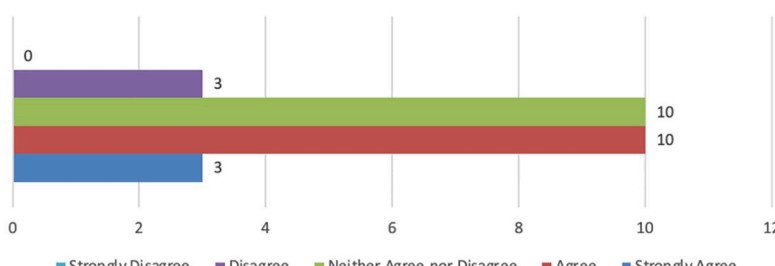

**Fig 22. Participants' views on Grad-CAM colouring scheme.** The figure shows that thirteen participants indicated that the colored heatmaps had a negative impact on the readability of the XAI results.

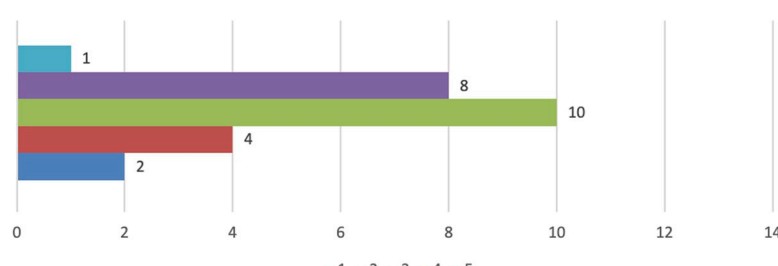

**Fig 23. LIME clinical relevance (Usefulness).** The figure shows that nine participants scored LIME less than 2 for the usefulness criteria.

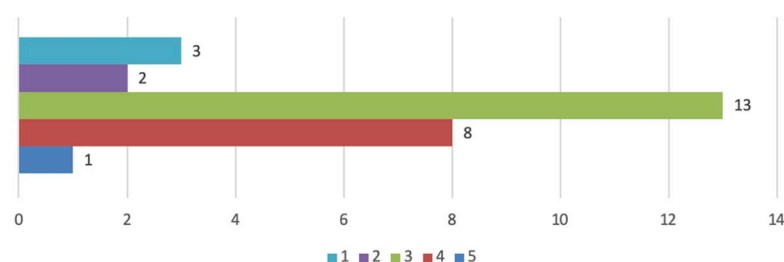

**Fig 24. Grad-CAM comprehensibility.** The figure shows that twenty-two participants rated the heatmap visualisations positively, with scores of three or higher.

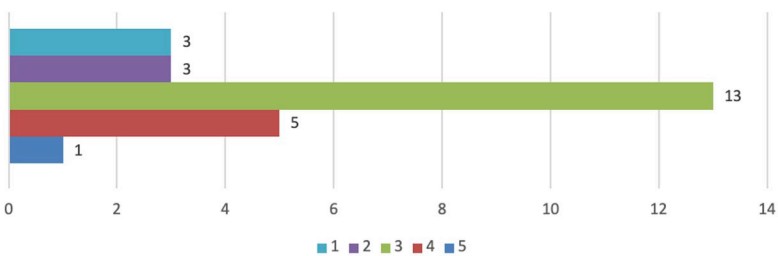

**Fig 25. LIME comprehensibility.** The figure shows that only six participants assigned a score of 4 or 5 for the comprehensibility criteria.

explanations and were easy to understand. Participants assigning lower scores provided varied justifications, including "Great explanation and detail." On the other hand, five participants rated the heatmap visualisations below three, primarily due to incorrect categorization in the heatmap, a loss of focus on the original image, and difficulty comprehending the heatmaps due to their unfamiliarity with them.

Regarding LIME visualizations, unlike Grad-CAM, where 9 participants provided rating of 4 or 5, only 6 participants assigned a score of 4 or 5 for the comprehensibility criteria (see Fig 25). On the other hand, six participants rated the visualisations below three, primarily because some unimportant areas were inaccurately highlighted, and the greyed-out areas introduced cognitive bias. Participants assigning lower scores provided varied justifications, including "I am more familiar with the basic image and is easier to recognize to my eyes."

In terms of the understandability and preference between the two XAI methods, nineteen participants favoured Grad-CAM (heatmap) over LIME visualisations, which were preferred by six participants (see Fig 26).

**5.2.3 Confidence.** When asked about any challenging or confusing aspects in the presentations of both XAI tools, participants had diverse feedback. Some participants suggested that the visualisations should include a confidence level for the classified regions of interest in diagnosis explanations. Others mentioned difficulties related to colour perception due to colour blindness, concerns about cognitive bias, and a desire for further explanations on the classification of areas. Participants provided further comments on this question such as, "*Scales should be provided for the healthy and disease segments.*"

In terms of confidence in the output from both XAI tools, nine participants expressed confidence in the accuracy of the visualisations, while seven participants lacked confidence in the results. Fig 27 illustrates the distribution of responses for the comparative evaluations and confidence in the XAI tools.

When queried about the influence of explanations provided by the tool on their trust in AI-based diagnostic systems, three participants indicated that their trust remained unchanged. They emphasized the ongoing necessity for physician supervision and accompanying reports

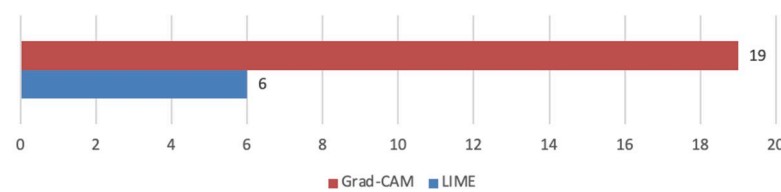

**Fig 26. Participants' preference between Grad-CAM and LIME.** The figure shows that nineteen participants favoured Grad-CAM (heatmap) over LIME visualisations.

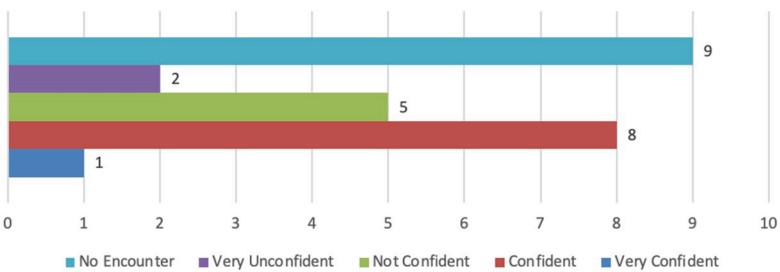

**Fig 27. Grad-CAM and LIME confidence.** The figure shows that nine participants expressed confidence in the accuracy of the XAI visualisations, while seven participants lacked confidence in the results.

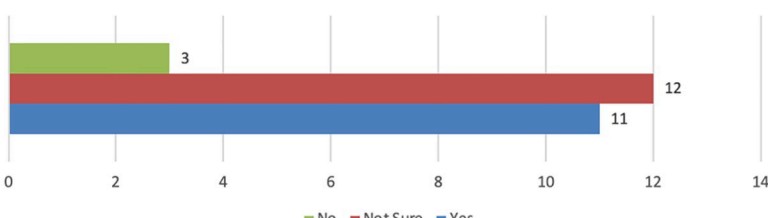

**Fig 28. Impact of XAI on improving trust in AI.** The figure shows that twelve participants expressed uncertainty about the impact on their trust in AI results in medical imaging, and eleven participants reported an improvement in their trust in AI systems after reviewing XAI visualizations.

alongside AI system outputs (see Fig 28). Meanwhile, twelve participants expressed uncertainty about the impact on their trust, attributing it to inaccuracies and insufficient detail in the provided visualizations. Some participants also highlighted their lack of AI knowledge as a factor affecting their confidence in the system outputs. Conversely, eleven participants reported an improvement in their trust in AI systems after reviewing XAI visualizations. When asked to justify their responses, diverse perspectives emerged. For instance, one participant emphasized the need for more engagement and medical training for the average clinician to develop trust in the additional value offered by these tools.

### 5.3 Part 4 results: Future recommendations

In this part, participants were asked about potential enhancements and recommendations to make XAI systems more effective and easily understandable for medical imaging tasks. Several interesting suggestions were proposed including the addition of commentary explaining to why specific areas are highlighted or the inclusion of a diagnostic report. Some participants also proposed the use of a different colour scheme for visualisations.

With regards to adding descriptive sentences (textual explanations) alongside visual explanations to improve clarity, one participant responded 'No', nine participants were uncertain, and fifteen participants responded 'Yes'. The main rationale for those in favour was that descriptions would assist in verifying the system's AI decision using a human language that is more coherent (see Fig 29).

Regarding the involvement of medical practitioners in the design and development of XAI systems for medical imaging, four participants neither agreed nor disagreed. They noted that factors such as time and the schedules of clinicians could affect their involvement in such projects. On the other hand, twenty-one participants either agreed or strongly agreed, emphasising

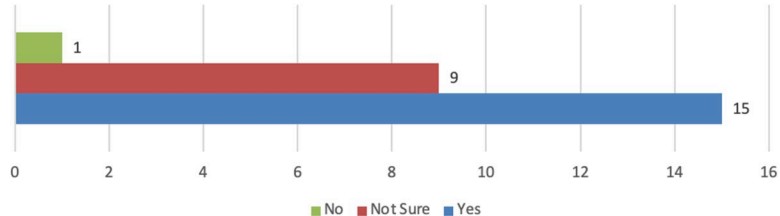

**Fig 29. Participants' support for textual explanations.** The figure shows that most participants support including textual explanations alongside the visual explanation for improved XAI results.

that clinicians are best suited to define their needs from medical imaging output and should prioritise the development of AI systems to meet those needs. As illustrated in Fig 30, some participants expressed that the involvement of clinicians would boost their confidence in the output of AI systems, and others highlighted the increasing integration of AI technologies into medical practices, underscoring the importance of clinicians' involvement in their development to ensure they meet the necessary standards.

Finally, Fig 31 illustrates that nineteen participants support the notion that XAI visualisations have the potential to enhance radiology practices. Meanwhile, four participants neither agreed nor disagreed, and one participant strongly disagreed. Those who expressed agreement believed that enhanced XAI visualisations could alleviate the workload on radiology services, aid in triaging patients for urgent care, improve the efficiency of medical radiology practices, and provide valuable assistance to clinicians. In contrast, the participant who disagreed indicated that the models were ineffective, and the XAI visualisations did not provide any help.

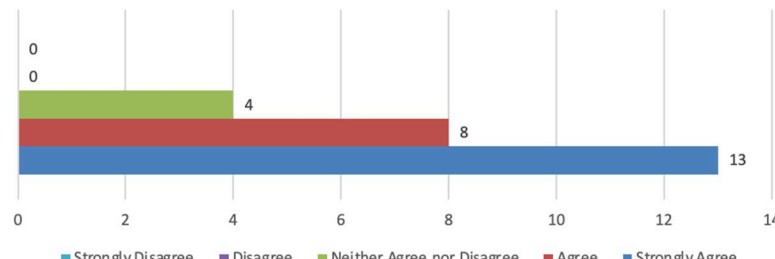

**Fig 30. Participants' support for involvement in XAI design and development.** The figure shows that most participants support XAI co-design.

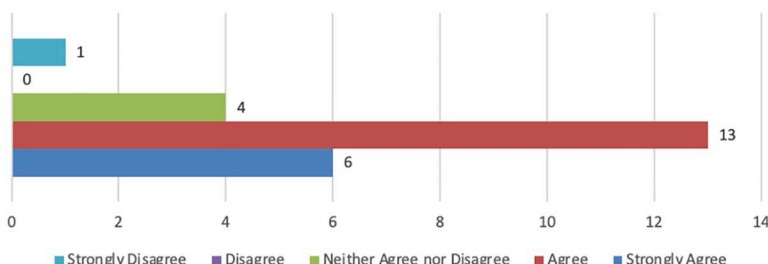

**Fig 31. Participants' belief in XAI benefits.** The figure illustrates that most participants believe that XAI visualisations have the potential to enhance radiology practices.

## 6 Discussion

Unlike in AI-based diagnostic systems, which uses standard performance evaluation measures (accuracy and F1 score), evaluation of XAI systems is not a standard practice in medical image analysis. Furthermore, in medicine a good explanation can differ between areas of expertise of the person for whom the explanation is given. With regards to chest radiology imaging, a good visual explanation should pinpoint to where disease is located in the image and the extent to the spread and impact on the pathology of the lung. Several visual XAI techniques have been proposed in the literature to explain AI outputs in chest radiology image analysis. However, little attention was paid to human grounded evaluation of such systems from the end users' perspective using practical clinical scenarios in chest radiology. This paper fills in this gap by designing a user study that evaluates the performance of XAI systems against a set of clinical criterion, including clinical relevance, comprehensibility and confidence, inline with the guidlines proposed in [24].

In assessing Grad-CAM and LIME XAI visualizations, our user study results reveal that, in terms of clinical relevance and comprehensibility, both XAI tools demonstrated an equally satisfactory performance. However, most participants preferred Grad-CAM heatmap visualizations over LIME, provided there is enhanced accuracy and additional details accompanying the visualizations to shed light on why the AI system identified certain areas as significant. The statement regarding whether trust in AI systems had improved with the use of XAI tools received the most neutral scores, which indicates that clinicians are most indifferent about this item.

Our findings draw attention to some key open problems, including:

- **The need to increase awareness of what XAI systems entails in chest radiology imaging**. Based on the survey findings, a majority of participants, despite possessing significant medical expertise, exhibited limited awareness of AI and XAI tools in medical imaging. This lack of familiarity can be attributed to little exposure to such tools in clinical settings and in current medical education. In addition, participants emphasized that enhanced knowledge of AI and XAI tool functionalities would boost their confidence levels. For example, there is a growing need to integrate AI education into medical training programs to facilitate better communication between these two fields.

- **The need for adopting user friendly multi-modal explanation, including both textual and visual explanations**. Combining visual explainability with textual description to obtain more coherent and human interpretable explanations is rarely considered in the chest radiology imaging domain. While there are some attempts to combine visual relevance and textual explanations (e.g [38]), these techniques were not designed with the user in mind and hence don't address the user's preferences and needs. To facilitate user interaction, the adoption of a user-friendly graphical interface designed for clinicians is also crucial.

- **The need for engaging key stakeholders (medical practitioners) in XAI design and development using real-life clinical scenarios**. The evolution of XAI in medical imaging is anticipated to incorporate substantial domain knowledge. To realize this vision, the active participation of key stakeholders, specifically medical experts, is essential throughout the entire life cycle of XAI development process. Collaborative efforts among physicians, AI researchers, and medical imaging experts will be a pivotal pathway for advancing the future development of AI-based methods in the area of medical imaging. Additionally, incorporating humans' perception in the design process is key to meet the explainability expectations and usability needs of XAI systems.

To advance these objectives in future research endeavors, several considerations are proposed. Firstly, there is a need for fine-tuning and enhancing deep learning diagnostic models to boost accuracy in identifying contributing factors to diagnostic predictions, ensuring more precise visual explanations when coupled with XAI tools. Second, specifying requirements for the clinical scenario in which the XAI technique is applied becomes essential to guarantee outputs align with clinicians' expectations. Furthermore, determining the level of detail to include in XAI visualizations, both in terms of regions of interest and textual diagnostic reports, is identified as a critical aspect to meet the specific needs of clinicians. Emphasizing transparency, it is suggested to report the confidence measures of XAI models in presented explanations, aiming to mitigate cognitive bias and enhance trust among clinicians.

Even though this study was carried out in the field of chest radiology imaging, the above points are transferable to other medical imaging modality (such as pathology and MRI images). Addressing the above concerns would significantly improve the quality of XAI systems, and more importantly increase the confidence, use and acceptance of XAI tools chest radiology practices.

The main limitation of this study is the few number of participants due to the scarcity of clinicians' availability and the low adoption of AI technology in the current radiology workflow.

## 7 Conclusion and future work

This paper evaluates two popular XAI systems in chest radiology imaging using human-centered evaluation method (user study), which involved 26 medical experts. To this end, we utilised pre-trained deep learning CNN-based models for diagnosing pneumonia and COVID-19 using open access chest X-rays and CT scans, respectively. Then, we applied Grad-CAM and LIME explainability methods, aiming to generate visual explanations for intended users (clinicians). Consequently, we conducted a qualitative user study to evaluate the visual explanation from the end users' perspective against a set of clinical criterion, including clinical relevance, comprehensibility and confidence. As far as we are aware, evaluating Grad-CAM and LIME explainability from the human perspective has not been explored previously in the context of chest radiology imaging.

In our study assessing Grad-CAM and LIME XAI visualizations, we found that both tools performed reasonably well in terms of clinical relevance and understanding. Interestingly, most participants favored Grad-CAM heatmaps over LIME, but they emphasized the importance of improved accuracy and more details to explain why the AI system identified certain areas as significant. When it came to the question of whether trust in AI systems had improved with the use of XAI tools, the responses were rather neutral. This suggests that clinicians seem to be indifferent or uncertain about whether these tools contribute significantly to building trust in AI systems.

Furthermore, our findings illuminate crucial factors essential for advancing XAI systems in chest radiology imaging. This involves raising awareness among medical experts about the potential benefits and mechanisms of AI and XAI in radiology imaging. There is a necessity to integrate user-friendly multi-modal explanations into current visual XAI systems to improve coherency. Additionally, actively involving clinicians in the design and development of XAI tools is crucial to create systems that are wll aligned with their specific needs.

## Supporting information

**S1 File. Developed software code for explainable deep learning model for pneumonia detection using chest X-ray images.** Colaboratory Python code for clinical case study 1—using Chest X-ray Images [33] The code can be accessed from https://colab.research.google.

com/drive/1v7RSS-_Prgujr-BrAGeDR_vygX_Tf-7r?usp=sharing here.
(ZIP)

**S2 File. Developed software code for explainable deep learning model for COVID-19 detection using chest CT images.** Colaboratory Python code for clinical case study 2—using Chest CT Images [35]. The code can be accessed from https://colab.research.google.com/drive/1Y1wjd9-sKLD6MaZDw4QVleSfAV22Ldb4?usp=sharing here.
(ZIP)

## Acknowledgments

The authors would like to thank all the anonymous participants for their time and valuable contributions to our study.

## Author Contributions

**Conceptualization:** Izegbua E. Ihongbe, Shereen Fouad, Bahadar Bhatia.

**Data curation:** Izegbua E. Ihongbe, Shereen Fouad.

**Formal analysis:** Izegbua E. Ihongbe, Shereen Fouad, Taha F. Mahmoud, Bahadar Bhatia.

**Funding acquisition:** Shereen Fouad.

**Investigation:** Izegbua E. Ihongbe, Shereen Fouad, Taha F. Mahmoud.

**Methodology:** Izegbua E. Ihongbe, Shereen Fouad, Taha F. Mahmoud.

**Project administration:** Shereen Fouad.

**Resources:** Izegbua E. Ihongbe, Shereen Fouad.

**Software:** Izegbua E. Ihongbe, Shereen Fouad.

**Supervision:** Shereen Fouad.

**Validation:** Izegbua E. Ihongbe, Shereen Fouad, Arvind Rajasekaran, Bahadar Bhatia.

**Visualization:** Izegbua E. Ihongbe, Shereen Fouad.

**Writing – original draft:** Izegbua E. Ihongbe, Shereen Fouad.

**Writing – review & editing:** Izegbua E. Ihongbe, Shereen Fouad, Taha F. Mahmoud, Arvind Rajasekaran, Bahadar Bhatia.

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
