## [Decision Letter · Decision Letter 0]

4 Jun 2024

PONE-D-24-03792Evaluating Explainable Artificial Intelligence (XAI) Techniques in Chest Radiology Imaging Through a human-centered LensPLOS ONE

Dear Dr. Fouad,

Thank you for submitting your manuscript to PLOS ONE. After careful consideration, we feel that it has merit but does not fully meet PLOS ONE’s publication criteria as it currently stands. Therefore, we invite you to submit a revised version of the manuscript that addresses the points raised during the review process.

We look forward to receiving your revised manuscript.

Kind regards,

Cosimo Ieracitano

Academic Editor

PLOS ONE

Journal Requirements:

4. We note that Figure(s) 3 and 4 in your submission contain copyrighted images. All PLOS content is published under the Creative Commons Attribution License (CC BY 4.0), which means that the manuscript, images, and Supporting Information files will be freely available online, and any third party is permitted to access, download, copy, distribute, and use these materials in any way, even commercially, with proper attribution. For more information, see our copyright guidelines: http://journals.plos.org/plosone/s/licenses-and-copyright.

a. You may seek permission from the original copyright holder of Figure(s) 3 and 4 to publish the content specifically under the CC BY 4.0 license. 

5. Kindly upload a separate folder for figure(s) 1 to 26. Please amend the file type to 'Figures'.

Additional Editor Comments:

AE: This manuscript has some merit, however it is not in a fine shape for consideration of acceptance in its current shape. Beside addressing the reviewers comments, the authors are asked to further motivate the chose of the xAI method used. In addition, the importance of xAI in every application domain (such as space [1] and intrusion detection [1]) should be discussed. [1]https://doi.org/10.1016/j.engappai.2024.108517. [2]https://doi.org/10.1016/j.eswa.2023.121751

Reviewers' comments:

Reviewer's Responses to Questions

**Comments to the Author**

1. Is the manuscript technically sound, and do the data support the conclusions?

Reviewer #1: Yes

Reviewer #2: Yes

2. Has the statistical analysis been performed appropriately and rigorously? 

Reviewer #1: Yes

Reviewer #2: N/A

3. Have the authors made all data underlying the findings in their manuscript fully available?

Reviewer #1: Yes

Reviewer #2: Yes

4. Is the manuscript presented in an intelligible fashion and written in standard English?

Reviewer #1: Yes

Reviewer #2: Yes

5. Review Comments to the Author

Reviewer #1: The authors proposed XAI model to predict the status of COVID-19.It is DL-enabled diagnostic systems in chest radiography. Two prominent XAI methods, Grad-CAM and LIME, are employed to generate visual explanations of the AI decision-making process. Two clinical scenarios for diagnosing pneumonia and COVID-19 using DL techniques are evaluated, achieving accuracy rates of 90% for pneumonia and 98% for COVID-19. The model seems interesting and may gain many interests, However, I have minor suggestions:

-The authors msy highlight recent XAI-Covid models. I suggest to highlight PMID:36738712 and similar methods.

-AUCROC plot must be drawn with mulitiple running points.

-how the authors checked whether the model overfits or not.

Reviewer #2: The paper analyzes the usefulness of xAI techniques (particularly Grad-CAM and LIME) in chest radiology (X-ray and CT) through the lens of the medical professionals who would leverage such advancements in a clinical context. Such an approach is critical in evaluating xAI techniques.

The study suggests that Grad-CAM was preferred over LIME regarding coherency and trust, and medical professionals are not aware of the potential uses of xAI.

Overall, the paper clearly identifies the gap that it aims to address (human-centered evaluation of Grad-CAM and LIME on a real use case of CAD), uses a grounded approach, and extracts reasonable conclusions from the results. However, the paper could benefit from a more detailed explanation of why Grad-CAM and LIME were the chosen xAI techniques when there are many others available (for example, SHAP is even mentioned in the literature review). It may also benefit from showing the performance of the discarded CNN architectures so one can better understand the weight of each evaluation metric in determining the model’s overall performance. As of now the influence of each metric seems arbitrarily defined.

6. PLOS authors have the option to publish the peer review history of their article (what does this mean?). If published, this will include your full peer review and any attached files.

Reviewer #1: **Yes: **ABEDALRHMAN

Reviewer #2: **Yes: **José Paulo Marques dos Santos

---

## [Author Response · Author response to Decision Letter 0]

8 Jul 2024

Response to Reviewers

Dear Reviewers,

We sincerely appreciate the time and effort you and the reviewers have dedicated to evaluating our paper and offering valuable feedback. Your insightful comments have significantly contributed to the improvements in this revised version. We have carefully considered each suggestion and endeavoured to address them thoroughly. We hope the revised manuscript meets your high standards and we welcome any further constructive feedback.

Below, we provide our point-by-point responses. As requested, all modifications in the marked-up manuscript have been highlighted in yellow.

Reviewers’ comments (highlighted in black text below) have been addressed in the updated manuscript and our response is provided in the blue text below. 

Sincerely,

Shereen Fouad, PhD, SFHEA

s.fouad@aston.ac.uk

‪Senior Lecturer in Computer Science‬‬‬‬‬‬‬‬‬‬‬‬‬

Aston University (College of Engineering and Physical Sciences)‬

Response to Reviewer 1

General Comment - The authors proposed XAI model to predict the status of COVID-19.It is DL-enabled diagnostic systems in chest radiography. Two prominent XAI methods, Grad-CAM and LIME, are employed to generate visual explanations of the AI decision-making process. Two clinical scenarios for diagnosing pneumonia and COVID-19 using DL techniques are evaluated, achieving accuracy rates of 90% for pneumonia and 98% for COVID-19. The model seems interesting and may gain many interests, However, I have minor suggestions:

Response – Thank you very much for your detailed review of our manuscript and your positive assessment. Your useful suggestions have been addressed below and included in our manuscript as advised.

Suggestion 1 - The authors msy highlight recent XAI-Covid models. I suggest to highlight PMID:36738712 and similar methods.

Response – Thank you for your suggestion, we highlighted recent XAI-Covid models by briefly discussing and citing the recommended article (reference 27) in section 2 (Literature review) paragraph 1, and the new text is highlighted in yellow in the updated manuscript.

Suggestion 2 - AUCROC plot must be drawn with mulitiple running points.

Response – Thank you for your suggestion, AUCROC plots have been provided for both datasets 1 and 2 (clinical case studies) in Figure 3 (3a and 3b), respectively. The figures are explained in section 3.2.2 paragraph 2, and the new text is highlighted in yellow in the updated manuscript.

Suggestion 3 - how the authors checked whether the model overfits or not.

Response – Thank you for your question. We implemented several techniques throughout our study to monitor and mitigate overfitting. In particular, our overfitting mitigation approach have been explained in the updated manuscript in section 3.2.1 (Experimental Settings) and the new text is highlighted in yellow in the updated manuscript.

We mentioned the followings: “We implemented several techniques throughout our study to monitor and mitigate overfitting. This includes applying regularization techniques, specifically L2 regularization and dropout to penalize model complexity and minimizing the risk of overfitting. For instance, a dropout rate of $1^{-0.5}$ was used in the MobileNetV2 and DenseNet169 models (best performing) to classify chest X-ray images and CT scans into pneumonia and normal, and COVID-19 and Non-COVID-19 cases, respectively. We also utilised early stopping in the models to stop training the model after its optimal number of iterations has been reached. Furthermore, both the training and validation loss curves were continuously monitored to ensure that no significant divergence between these curves occurred, which is often a good indicator of overfitting.”

Response to Reviewer 2

General Comment - The paper analyzes the usefulness of xAI techniques (particularly Grad-CAM and LIME) in chest radiology (X-ray and CT) through the lens of the medical professionals who would leverage such advancements in a clinical context. Such an approach is critical in evaluating xAI techniques.

The study suggests that Grad-CAM was preferred over LIME regarding coherency and trust, and medical professionals are not aware of the potential uses of xAI.

Overall, the paper clearly identifies the gap that it aims to address (human-centered evaluation of Grad-CAM and LIME on a real use case of CAD), uses a grounded approach, and extracts reasonable conclusions from the results.

Response – Thank you very much for your thorough review of our manuscript and your thoughtful evaluation. Your useful suggestions have been addressed below and included in our manuscript as advised.

Suggestion 1 - However, the paper could benefit from a more detailed explanation of why Grad-CAM and LIME were the chosen xAI techniques when there are many others available (for example, SHAP is even mentioned in the literature review). 

Response – we included a detailed discussion in section (3.3 Visual Explainability models) justifying why Grad-CAM and LIME were the chosen xAI techniques in our study. and the new text is highlighted in yellow in the updated manuscript.

We explained that: 

“Based on our initial experimental findings, Grad-CAM [12] and LIME [13] provide more stable and accurate localized explanations compared to SHAP [14] in both image classification tasks. Therefore, in this paper, we selected LIME and Grad-CAM methods due to their superior performance in delivering accurate, relevant, and stable explainability results. Evidence from recent literature supports this choice. For instance, a study in remote sensing image classification [35] compared the performance of ten different XAI methods and found that Grad-CAM and LIME were the most interpretable and reliable. Similarly, a research in [16] comparing Grad-CAM, SHAP, and LIME in the context of medical imaging concluded that Grad-CAM and LIME were more reliable, whereas SHAP was not the best for local accuracy in this application. This is consistent with findings in [14], which highlights that while SHAP provides comprehensive feature importance in non-imaging datasets, it may produce less stable explanations in complex image classification tasks, leading to potential inconsistencies. These findings underscore the reliability and relevance of LIME and Grad-CAM in our study, facilitating better insights and trust in the model outputs. “

The above text was included in section 3.3 pages 6&7, and it is highlighted in yellow in the updated manuscript.

Suggestion 2 - It may also benefit from showing the performance of the discarded CNN architectures so one can better understand the weight of each evaluation metric in determining the model’s overall performance. As of now the influence of each metric seems arbitrarily defined.

Response – Thank you for your suggestion, we have included the experimental results obtained from all the studied deep learning models for dataset 1 and 2 in Tables 1 and 2, respectively. The test results are reported in section 3.2.2 (Results). The Performance Metrics (on testset) of Deep Learning Models on Dataset 1 and 2 are reported in terms of Accuracy, Precision, Recall, and F1 Score. We also report the training and validation loss and accuracy plots for the best performing models, in Dataset 1 and 2, in Figures 1 and 2, respectively. The updated text is highlighted in yellow in the updated manuscript.

---

## [Decision Letter · Decision Letter 1]

30 Jul 2024

Evaluating Explainable Artificial Intelligence (XAI) Techniques in Chest Radiology Imaging Through a human-centered Lens

PONE-D-24-03792R1

Dear Dr. Fouad,

We’re pleased to inform you that your manuscript has been judged scientifically suitable for publication and will be formally accepted for publication once it meets all outstanding technical requirements.

Kind regards,

Cosimo Ieracitano

Academic Editor

PLOS ONE

Additional Editor Comments (optional):

Reviewers' comments:

Reviewer's Responses to Questions

**Comments to the Author**

1. If the authors have adequately addressed your comments raised in a previous round of review and you feel that this manuscript is now acceptable for publication, you may indicate that here to bypass the “Comments to the Author” section, enter your conflict of interest statement in the “Confidential to Editor” section, and submit your "Accept" recommendation.

Reviewer #1: All comments have been addressed

Reviewer #2: All comments have been addressed

2. Is the manuscript technically sound, and do the data support the conclusions?

Reviewer #1: Yes

Reviewer #2: Yes

3. Has the statistical analysis been performed appropriately and rigorously? 

Reviewer #1: Yes

Reviewer #2: N/A

4. Have the authors made all data underlying the findings in their manuscript fully available?

Reviewer #1: (No Response)

Reviewer #2: Yes

5. Is the manuscript presented in an intelligible fashion and written in standard English?

Reviewer #1: Yes

Reviewer #2: Yes

6. Review Comments to the Author

Reviewer #1: The authors have addressed the reviewers comments adequately. The manuscript is in a very good shape.

Reviewer #2: The authors addressed all the previous comments/suggestions. Therefore, I recommend the article for publication.

7. PLOS authors have the option to publish the peer review history of their article (what does this mean?). If published, this will include your full peer review and any attached files.

Reviewer #1: **Yes: **Abedalrhman Alkhateeb

Reviewer #2: **Yes: **José Paulo Marques dos Santos

---

## [Editor Report · Acceptance letter]

14 Aug 2024

PONE-D-24-03792R1 

PLOS ONE

Dear Dr. Fouad, 

I'm pleased to inform you that your manuscript has been deemed suitable for publication in PLOS ONE. Congratulations! Your manuscript is now being handed over to our production team.

Kind regards, 

on behalf of

Dr. Cosimo Ieracitano 

Academic Editor

PLOS ONE